# Learning Programmatically Structured Representations with Perceptor Gradients

**Svetlin Penkov** [1,2,*], **Subramanian Ramamoorthy** [1,2]
[1]The University of Edinburgh, [2]FiveAI
{sv.penkov, s.ramamoorthy}@ed.ac.uk

## Abstract

We present the perceptor gradients algorithm – a novel approach to learning symbolic representations based on the idea of decomposing an agent's policy into i) a perceptor network extracting symbols from raw observation data and ii) a task encoding program which maps the input symbols to output actions. We show that the proposed algorithm is able to learn representations that can be directly fed into a Linear-Quadratic Regulator (LQR) or a general purpose A* planner. Our experimental results confirm that the perceptor gradients algorithm is able to efficiently learn transferable symbolic representations as well as generate new observations according to a semantically meaningful specification.

## 1 Introduction

Learning representations that are useful for devising autonomous agent policies to act, from raw data, remains a major challenge. Despite the long history of work in this area, fully satisfactory solutions are yet to be found for situations where the representation is required to address symbols whose specific meaning might be crucial for subsequent planning and control policies to be effective. So, on the one hand, purely data-driven neural network based models, e.g., (Duan et al., 2016; Arulkumaran et al., 2017), make significant data demands and also tend to overfit to the observed data distribution (Srivastava et al., 2014). On the other hand, symbolic methods that offer powerful abstraction capabilities, e.g., (Kansky et al., 2017; Verma et al., 2018), are not yet able to learn from raw noisy data. Bridging this gap between neural and symbolic approaches is an area of active research (Garnelo et al., 2016; Garcez et al., 2018; Kser et al., 2017; Penkov & Ramamoorthy, 2017).

In many applications of interest, the agent is able to obtain a coarse 'sketch' of the solution quite independently of the detailed representations required to execute control actions. For instance, a human user might be able to provide a programmatic task description in terms of abstract symbols much easier than more detailed labelling of large corpora of data. In the literature, this idea is known as end-user programming (Lieberman et al., 2006), programming by demonstration (Billard et al., 2008) or program synthesis (Gulwani et al., 2017). In this space, we explore the specific hypothesis that a programmatic task description provides inductive bias that enables significantly more efficient learning of symbolic representations from raw data. The symbolic representation carries semantic content that can be grounded to objects, relations or, in general, any pattern of interest in the environment.

We address this problem by introducing the *perceptor gradients* algorithm which decomposes a typical policy, mapping from observations to actions, into i) a *perceptor* network that maps observations to symbolic representations and ii) a user-provided task encoding *program* which is executed on the perceived symbols in order to generate an action. We consider both feedforward and autoencoding perceptors and view the program as a regulariser on the latent space which not only provides a strong inductive bias structuring the latent space, but also attaches a semantic meaning to the learnt representations. We show that the perceptor network can be trained using the REINFROCE estimator (Williams, 1992) for *any* task encoding program.

We apply the perceptor gradients algorithm to the problem of balancing a cart-pole system with a Linear-Quadratic Regulator (LQR) from pixel observations, showing that the state variables over

---

*Work done as part of the author's PhD thesis at the University of Edinburgh.

which a concise control law could be defined is learned from data. Then, we demonstrate the use of the algorithm in a navigation and search task in a Minecraft-like environment, working with a 2.5D rendering of the world, showing that symbols for use within a generic A* planner can be learned. We demonstrate that the proposed algorithm is not only able to efficiently learn transferable symbolic representations, but also enables the generation of new observations according to a semantically meaningful specification. We examine the learnt representations and show that programmatic regularisation is a general technique for imposing an inductive bias on the learning procedure with capabilities beyond the statistical constraints typically used.

## 2 RELATED WORK

Representation learning approaches rely predominantly on imposing statistical constraints on the latent space (Bengio et al., 2013) such as minimising predictability (Schmidhuber, 1992), maximising independence (Barlow et al., 1989; Higgins et al., 2017), minimising total correlation (Chen et al., 2018; Kim & Mnih, 2018) or imposing structured priors on the latent space (Chen et al., 2016; Narayanaswamy et al., 2017). While learning disentangled features is an important problem, it does not by itself produce features of direct relevance to the planning/control task at hand. For example, the 'best' features describing a dynamic system may be naturally entangled, as in a set of differential equations. A programmatic task representation allows such dependencies to be expressed, making subsequent learning more efficient and improving generalisation capabilities (Kansky et al., 2017; Kusner et al., 2017; Gaunt et al., 2017; Verma et al., 2018).

Indeed, the idea of leveraging domain knowledge and imposing model driven constraints on the latent space, has been studied in different domains. Much of this work is focused on learning representations for predicting and controlling physical systems where various assumptions about the underlying dynamic model are made (Watter et al., 2015; Iten et al., 2018; Fraccaro et al., 2017; Karl et al., 2017). Bezenac et al. (2018) even leverage a theoretical fluid transport model to forecasting sea surface temperature. Model based representation learning approaches have also been applied to inverse graphics, where a visual renderer guides the learning process (Mansinghka et al., 2013; Kulkarni et al., 2014; Ellis et al., 2018), and inverse physics, where a physics simulator is utilised (Wu et al., 2017). Interestingly, Kulkarni et al. (2015) propose a renderer that takes a programmatic description of the scene as input, similar to the one used in Ellis et al. (2018). Of particular relevance to our work is the idea of learning compositional visual concepts by enforcing the support of set theoretic operations such as union and intersection in the latent space (Higgins et al., 2018). Programmatic regularisation, which we propose, can be viewed as a general tool for expressing such model based constraints.

Programmatic task descriptions can be obtained through natural language instructions (Kaplan et al., 2017; Matuszek et al., 2013), demonstrations in virtual environments (Penkov & Ramamoorthy, 2017) or even manually specified. Importantly, problems that appear quite hard from a pixel-level view can actually be more easily solved by constructing simple programs which utilise symbols that exploit the structure of the given problem. Sometimes, utilising prior knowledge in this way has been avoided in a puristic approach to representation and policy learning. In this paper, we address the more pragmatic view wherein the use of coarse task descriptions enables the autonomous agents to efficiently learn more abstract representations with semantic content that are relevant to practice.

## 3 PROBLEM DEFINITION

Let us consider the Markov decision process (MDP) represented as the tuple $(\mathcal{S}, \mathcal{A}, P, r, \gamma, P_0)$ where $\mathcal{S}$ is the set of possible states, $\mathcal{A}$ is the set of possible actions, $P$ is the state transition probability distribution, $r : \mathcal{S} \times \mathcal{A} \to \mathbb{R}$ is the reward function, $\gamma$ is the reward discounting factor and $P_0$ is the probability distribution over the initial states. At each time step $t$, the agent observes a state $\mathbf{s}_t \in \mathcal{S}$ and chooses an action $\mathbf{a}_t \in \mathcal{A}$ which results in a certain reward $r_t$. We consider the stochastic policy $\pi_\theta(\mathbf{a}_t | \mathbf{s}_t)$ belonging to a family of functions parameterised by $\theta$ such that $\pi_\theta : \mathcal{S} \to \mathcal{P}_\mathcal{A}(\mathcal{A})$ where $\mathcal{P}_\mathcal{A}$ is a probability measure over the set $\mathcal{A}$.

We are interested in decomposing the policy $\pi_\theta$ into i) a perceptor $\psi_\theta : \mathcal{S} \to \mathcal{P}_\Sigma(\Sigma)$, where $\Sigma$ is a set of task related symbols and $\mathcal{P}_\Sigma$ is a probability measure over it, and ii) a task encoding program

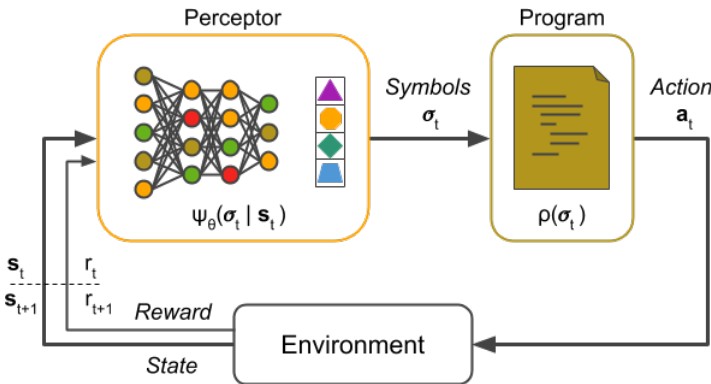

Figure 1: The proposed policy decomposition into a perceptor and a program.

$\rho : \Sigma \to \mathcal{A}$ such that $\rho \circ \psi_\theta : \mathcal{S} \to \mathcal{A}$ as shown in Figure 1. In this paper, we address the problem of learning $\psi_\theta$ with any program $\rho$ and investigate the properties of the learnt set of symbols $\Sigma$.

## 4 METHOD

Starting from an initial state of the environment $\mathbf{s}_0 \sim P_0(\mathbf{s})$ and following the policy $\pi_\theta$ for $T$ time steps results in a trace $\tau = (\mathbf{s}_0, \mathbf{a}_0, \mathbf{s}_1, \mathbf{a}_1, \ldots, \mathbf{s}_T, \mathbf{a}_T)$ of $T+1$ state-action pairs and $T$ reward values $(r_1, r_2, \ldots, r_T)$ where $r_t = r(\mathbf{s}_t, \mathbf{a}_t)$. The standard reinforcement learning objective is to find an optimal policy $\pi_\theta^*$ by searching over the parameter space, such that the total expected return

$$J(\theta) = E_{\tau \sim p(\tau;\theta)}\left[R_0(\tau)\right] = \int p(\tau;\theta) R_0(\tau)\, d\tau \tag{1}$$

is maximised, where $R_t(\tau) = \sum_{i=t}^T \gamma^{i-t} r(\mathbf{s}_t, \mathbf{a}_t)$ is the return value of state $\mathbf{s}_t$.

The REINFORCE estimator (Williams, 1992) approximates the gradient of the cost as

$$\nabla_\theta J(\theta) \approx \frac{1}{n} \sum_{i=1}^n \sum_{t=0}^{T-1} \nabla_\theta \log \pi_\theta(\mathbf{a}_t^{(i)}|\mathbf{s}_t^{(i)}) \left( R_t(\tau^{(i)}) - b_\phi(\mathbf{s}_t^{(i)}) \right) \tag{2}$$

where $n$ is the number of observed traces, $b_\phi$ is a baseline function paramterised by $\phi$ introduced to reduce the variance of the estimator.

Let us now consider the following factorisation of the policy

$$\pi_\theta(\mathbf{a}_t|\mathbf{s}_t) = p(\mathbf{a}_t|\boldsymbol{\sigma}_t)\psi_\theta(\boldsymbol{\sigma}_t|\mathbf{s}_t) \tag{3}$$

where $\boldsymbol{\sigma}_t \in \Sigma$ are the symbols extracted from the state $\mathbf{s}_t$ by the perceptor $\psi_\theta$, resulting in the augmented trace $\tau = (\mathbf{s}_0, \boldsymbol{\sigma}_0, \mathbf{a}_0, \mathbf{s}_1, \boldsymbol{\sigma}_1, \mathbf{a}_1, \ldots, \mathbf{s}_T, \boldsymbol{\sigma}_T, \mathbf{a}_T)$. We are interested in exploiting the programmatic structure supported by the symbols $\boldsymbol{\sigma}_t$ and so we set

$$p(\mathbf{a}_t|\boldsymbol{\sigma}_t) = \delta_{\rho(\boldsymbol{\sigma}_t)}(\mathbf{a}_t) \tag{4}$$

which is a Dirac delta distribution centered on the output of the task encoding program $\rho(\boldsymbol{\sigma}_t)$ for the input symbols $\boldsymbol{\sigma}_t$. Even though the program should produce a single action, it could internally work with distributions and simply sample its output. Decomposing the policy into a program and a perceptor enables the description of programmatically structured policies, while being able to learn the required symbolic representation from data. In order to learn the parameters of the perceptor $\psi_\theta$, we prove the following theorem.

**Theorem 1** (Perceptor Gradients). *For any decomposition of a policy $\pi_\theta$ into a program $\rho$ and a perceptor $\psi_\theta$ such that*

$$\pi_\theta(\mathbf{a}_t|\mathbf{s}_t) = \delta_{\rho(\boldsymbol{\sigma}_t)}(\mathbf{a}_t)\ \psi_\theta(\boldsymbol{\sigma}_t|\mathbf{s}_t) \tag{5}$$

*the gradient of the log-likelihood of a trace sample $\tau^{(i)}$ obtained by following $\pi_\theta$ is*

$$\nabla_\theta \log p(\tau^{(i)};\theta) = \sum_{t=0}^{T-1} \nabla_\theta \log \psi_\theta(\boldsymbol{\sigma}_t^{(i)}|\mathbf{s}_t^{(i)}) \tag{6}$$

**Proof.** Substituting $\pi_\theta(\mathbf{a}_t|\mathbf{s}_t)$ according to (5) gives

$$
\begin{aligned}
\nabla_\theta \log p(\tau^{(i)};\theta) = \sum_{t=0}^{T-1} \nabla_\theta \log \pi_\theta(\mathbf{a}_t^{(i)}|\mathbf{s}_t^{(i)}) = \\
= \sum_{t=0}^{T-1} \left[ \nabla_\theta \log \delta_{\rho(\boldsymbol{\sigma}_t^{(i)})}(\mathbf{a}_t^{(i)}) + \nabla_\theta \log \psi_\theta(\boldsymbol{\sigma}_t^{(i)}|\mathbf{s}_t^{(i)}) \right] = \\
= \sum_{t=0}^{T-1} \nabla_\theta \log \psi_\theta(\boldsymbol{\sigma}_t^{(i)}|\mathbf{s}_t^{(i)}) \qquad \blacksquare
\end{aligned}
$$

Theorem 1 has an important consequence – no matter what program $\rho$ we choose, as long as it outputs an action in a finite amount of time, the parameters $\theta$ of the perceptor $\psi_\theta$ can be learnt with the standard REINFORCE estimator.

**Feedforward Perceptor**    By combining theorem 1 with (2), we derive the following loss function

$$
\mathcal{L}(\theta,\phi) = \frac{1}{n} \sum_{i=1}^{n} \left\{ \mathcal{L}_\psi(\tau^{(i)},\theta) + \mathcal{L}_b(\tau^{(i)},\phi) \right\} \tag{7}
$$

$$
\mathcal{L}_\psi(\tau^{(i)},\theta) = \sum_{t=0}^{T-1} \log \psi_\theta(\boldsymbol{\sigma}_t^{(i)}|\mathbf{s}_t^{(i)}) \left( R_t(\tau^{(i)}) - b_\phi(\mathbf{s}_t^{(i)}) \right) \tag{8}
$$

$$
\mathcal{L}_b(\tau^{(i)},\phi) = \sum_{t=0}^{T-1} \left( R_t(\tau^{(i)}) - b_\phi(\mathbf{s}_t^{(i)}) \right)^2 \tag{9}
$$

and $\tau^{(i)} = (\mathbf{s}_0, \boldsymbol{\sigma}_0, \mathbf{s}_1, \boldsymbol{\sigma}_1, \ldots, \mathbf{s}_T, \boldsymbol{\sigma}_T)$ contains $T+1$ sampled state-symbol pairs. Algorithm 1 demonstrates how to rollout a policy decomposed into a perceptor and a program in order to obtain trajectory samples $\tau^{(i)}$, while the overall perceptor gradients learning procedure is summarised in Algorithm 2.

**Autoencoding Perceptor**    The perceptor is a mapping such that $\psi_\theta : \mathcal{S} \to \mathcal{P}_\Sigma(\Sigma)$ and so by learning the inverse mapping $\omega_\upsilon : \Sigma \to \mathcal{P}_\mathcal{S}(\mathcal{S})$, parameterised by $\upsilon$, we enable the generation of states (observations) from a structured symbolic description. Thus $\psi_\theta$ and $\omega_\upsilon$ form an autoencoder, where the latent space is $\Sigma$. Importantly, the resulting autoencoder can be trained efficiently by applying the reparameterisation trick (Kingma & Welling, 2014; Jang et al., 2016) when sampling values for $\boldsymbol{\sigma}_t$ during the perceptor rollout and reusing the obtained samples for the training of the generator $\omega_\upsilon$. In order to do so, we simply augment the loss in (7) with a reconstruction loss term

$$
\mathcal{L}_\omega(\tau^{(i)},\theta) = \sum_{t=0}^{T-1} \log \omega_\upsilon(\mathbf{s}_t^{(i)}|\boldsymbol{\sigma}_t^{(i)}) \tag{10}
$$

## 5    Experimental Results

### 5.1    Cart-Pole Balancing

We first consider the problem of balancing a cart-pole system by learning symbolic representations from the raw image observations. The cart-pole system is well studied in optimal control theory and it is typically balanced with an LQR (Zhou et al., 1996). We exploit this knowledge and set the program $\rho$ to implement an LQR. The perceptor $\psi_\theta$ is a convolutional neural network (see A.1) as shown in the overall experiment diagram in Figure 2. We define the state vector as

$$
\boldsymbol{\sigma} = \begin{bmatrix} x & \dot{x} & \alpha & \dot{\alpha} \end{bmatrix}^T \tag{11}
$$

where $x \in \mathbb{R}$ is the linear position of the cart and $\alpha \in \mathbb{R}$ is the angle of the pendulum with respect to its vertical position as shown in Figure 2.

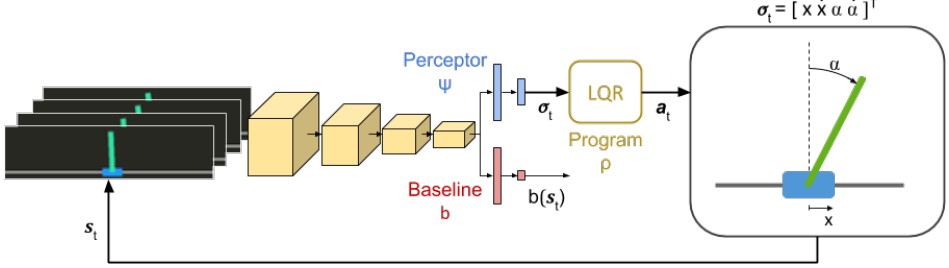

Figure 2: A diagram of the cart-pole experimental setup.

**LQR Program** Given the standard state space representation of a cart-pole system (see A.3) we can design a feedback control law $u = -\boldsymbol{K}\boldsymbol{\sigma}$, where $u$ corresponds to the force applied to the cart and $\boldsymbol{K}$ is a gain matrix. In an LQR design, the gain matrix $\boldsymbol{K}$ is found by minimising the quadratic cost function

$$J = \int_0^\infty \boldsymbol{\sigma}(t)^T \boldsymbol{Q}\boldsymbol{\sigma}(t) + u(t)^T \boldsymbol{R}u(t) \; dt \quad (12)$$

where we set $\boldsymbol{Q} = 10^3 \; \boldsymbol{I}_4$ and $R = 1$ as proposed in (Lam, 2004). We set the program $\rho$ to be

$$\rho(\boldsymbol{\sigma}) = \mathbf{a} = \begin{cases} 1 & \text{if } -\boldsymbol{K}\boldsymbol{\sigma} > 0 \\ 0 & \text{otherwise} \end{cases} \quad (13)$$

producing 2 discrete actions required by the OpenAI gym cart-pole environment. We used the `python-control`[1] package to estimate $\boldsymbol{K} = [-1 \; -2.25 \; -30.74 \; -7.07]$.

### 5.1.1 LEARNING PERFORMANCE

In this experimental setup the perceptor is able to learn from raw image observations the symbolic representations required by the LQR controller. The average reward obtained during training is shown in Figure 3. We compare the performance of the perceptor gradients algorithm to a standard policy gradients algorithm, where we have replaced the program with a single linear layer with sigmoid activation. The perceptor obtains an average reward close to the maximum of 199 approximately after 3000 iterations compared to 7000 iterations for the standard policy, however the obtained reward has greater variance. Intuitively this can be explained with the fact that the program encodes a significant amount of knowledge about the task which speeds up the learning, but also defines a much more constrained manifold for the latent space that is harder to be followed during stochastic gradient descent.

**Algorithm 1:** Perceptor rollout for a single episode

**Input:** $\psi_\theta, \rho$
**Output:** $\tau, r_{1:T}$

**for** $t = 0$ **to** $T$ **do**
  $\mathbf{s}_t \leftarrow$ observe environment
  $\boldsymbol{\sigma}_t \leftarrow$ sample from $\psi_\theta(\boldsymbol{\sigma}|\mathbf{s}_t)$
  $\mathbf{a}_t \leftarrow \rho(\boldsymbol{\sigma}_t)$
  $r_t \leftarrow$ execute $\mathbf{a}_t$
  append $(\mathbf{s}_t, \boldsymbol{\sigma}_t)$ to $\tau$

**Algorithm 2:** Perceptor gradients

$(\theta, \phi) \leftarrow$ Initialise parameters
**repeat**
  **for** $i = 1$ **to** $n$ **do**
    $\tau^{(i)}, r_{1:T}^{(i)} \leftarrow$ rollout$(\psi_\theta, \rho)$
    **for** $t = 0$ **to** $T$ **do**
      $R_t^{(i)} \leftarrow \sum_{i=t}^T \gamma^{i-t} r_t^{(i)}$
      $A_t^{(i)} \leftarrow \left( R_t^{(i)} - b_\phi(\mathbf{s}_t^{(i)}) \right)$
    $\mathcal{L}_\psi \leftarrow \frac{1}{n} \sum_{i=1}^n \sum_{t=0}^{T-1} \log \psi_\theta(\boldsymbol{\sigma}_t^{(i)}|\mathbf{s}_t^{(i)}) A_t^{(i)}$
    $\mathcal{L}_b \leftarrow \frac{1}{n} \sum_{i=1}^n \sum_{t=0}^{T-1} \left( R_t^{(i)} - b_\phi(\mathbf{s}_t^{(i)}) \right)^2$
    $\mathcal{L} \leftarrow \mathcal{L}_\psi + \mathcal{L}_b$
    $\mathbf{g} \leftarrow \nabla_{\theta, \phi} \mathcal{L}(\theta, \phi)$
  $(\theta, \phi) \leftarrow$ Update parameters using $\mathbf{g}$
**until** convergence of parameters $(\theta, \phi)$;

### 5.1.2 PERCEPTOR LATENT SPACE

The state space for which the minimum of the LQR cost in (12) is obtained is defined only up to scale and rotation (see A.4). Therefore, we use one episode of labelled data to find the underlying linear transformation through constrained optimisation. The transformed output of the perceptor for an entire episode is shown in Figure 4. The position and angle representations learnt by the perceptor

---

[1] http://python-control.org

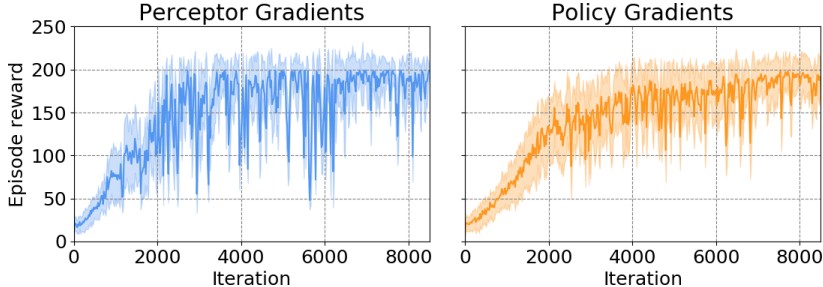

Figure 3: Learning performance at the cart-pole balancing task of the perceptor gradients algorithm (left) compared to standard policy gradients (right).

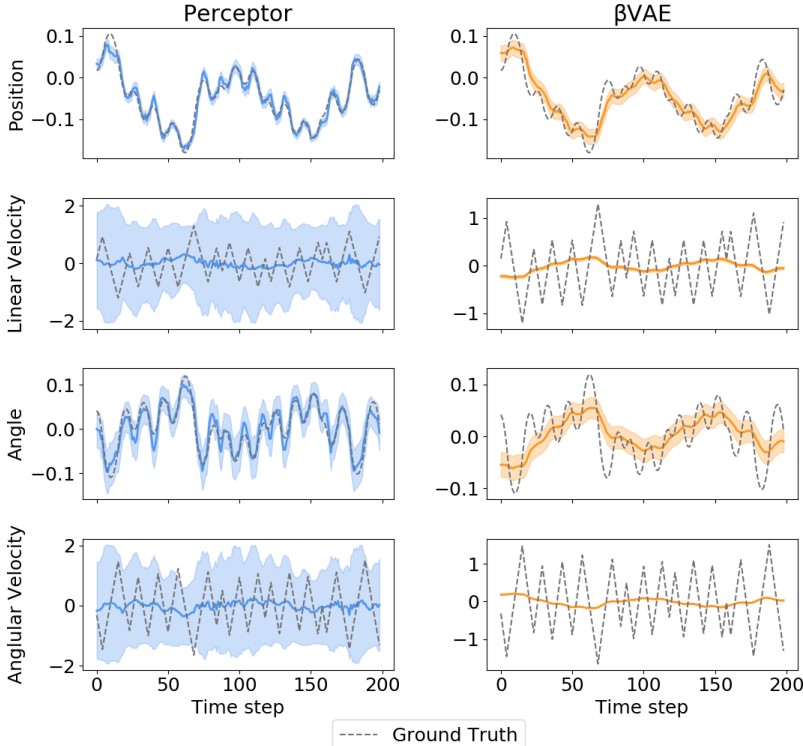

Figure 4: The latent space learnt by a perceptor (left) and $\beta$VAE (right). The thick coloured lines represent the predicted mean, while the shaded regions represent $\pm 1\sigma$ of the predicted variance.

match very closely the ground truth. However, both the linear and angular velocities do not match the true values and the associated variance is also quite large. Investigating the results, we found out that the LQR controller is able to balance the system even if the velocities are not taken into account. Given that the performance of the task is not sensitive to the velocity estimates, the perceptor was not able to ground the velocities from the symbolic state representation to the corresponding patterns in the observed images.

We compared the representations learnt by the perceptor to the ones learnt by a $\beta$VAE (Higgins et al., 2017). In order to do so, we generated a dataset of observations obtained by controlling the cart-pole with the perceptor for 100 episodes. We set the $\beta$VAE encoder to be architecturally the same as the perceptor and replaced the convolutional layers with transposed convolutions for the decoder. We performed linear regression between the latent space of the trained $\beta$VAE and the ground truth values. As it can be seen from the results in Figure 4 the representations learnt by the $\beta$VAE do capture some of the symbolic state structure, however they are considerably less precise

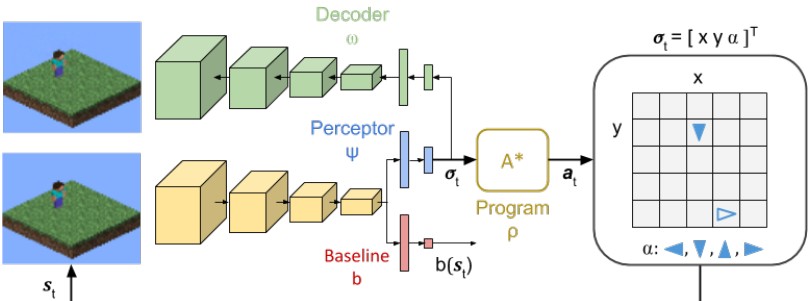

Figure 5: A diagram of the 'go to pose' experimental setup.

than the ones learnt by the perceptor. The $\beta$VAE does not manage to extract the velocities either and also collapses the distribution resulting in high certainty of the wrong estimates.

As proposed by Higgins et al. (2017), setting $\beta > 1$ encourages the $\beta$VAE to learn disentangled representations. The results shown in Figure 4, though, were obtained with $\beta = 0.1$, otherwise the regularisation on the latent space forcing the latent dimensions to be independent was too strong. It has been recognised by the community that there are better ways to enforce disentanglement such as minimising total correlation (Chen et al., 2018; Kim & Mnih, 2018). However, these methods also rely on the independence assumption which in our case does not hold. The position and the angle of the cart-pole system are clearly entangled and the program is able to correctly bias the learning procedure as it captures the underlying dependencies.

## 5.2 MINECRAFT: GO TO POSE

**Task Description**  We apply the perceptor gradients algorithm to the problem of navigating to a certain pose in the environment by learning symbolic representations from images. In particular, we consider a $5 \times 5$ grid world where an agent is to navigate from its location to a randomly chosen goal pose. The agent receives +1 reward if it gets closer to the selected goal pose, +5 if it reaches the position and +5 if it rotates to the right orientation. To encourage optimal paths, the agent also receives -0.5 reward at every timestep. A symbolic representation of the task together with the 2.5D rendering of the environment and the autoencoding perceptor, which we train (see A.2), are shown in Figure 5. We express the state of the environment as

$$\boldsymbol{\sigma} = \begin{bmatrix} x & y & \alpha \end{bmatrix}^T \qquad (14)$$

where $x, y \in \{1, 2, 3, 4, 5\}$ are categorical variables representing the position of the agent in the world and $\alpha \in \{1, 2, 3, 4\}$ represents its orientation.

**A* Program**  Given that the pose of the agent $\boldsymbol{\sigma}$ and the goal pose $G$ are known this problem can be easily solved using a general purpose planner. Therefore, in this experiment the program $\rho$ implements a general purpose A* planner. In comparison to the simple control law program we used in the cart-pole experiments, $\rho$ in this case is a much more complex program as it contains several loops, multiple conditional statements as well as a priority queue. For the experiments in this section we directly plugged in the implementation provided by the `python-astar` package [2].

At every timestep, a path is found between the current $\boldsymbol{\sigma}_t$ produced by the perceptor and the goal $G$ randomly chosen at the beginning of the episode such that the agent either moves to one of the 4 neighbouring squares or rotates in-place at $90°$, $180°$ or $270°$. The output action $\mathbf{a}_t = \rho(\boldsymbol{\sigma_t})$ is set to the first action in the path found by the A* planner.

### 5.2.1 LEARNING PERFORMANCE

In this experimental setup the perceptor is able to learn the symbolic representations required by the A* planner from raw image observations. The average reward obtained during training is shown in Figure 6. Again, we compare the performance of the perceptor gradients to a standard policy

---

[2]`https://pypi.org/project/astar/`

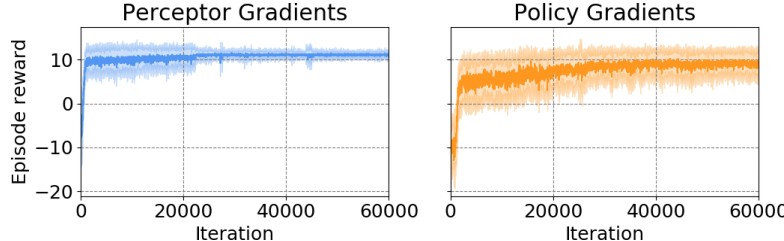

Figure 6: Learning performance at the 'go to pose' task of the perceptor gradients (left) compared to the policy gradients (right).

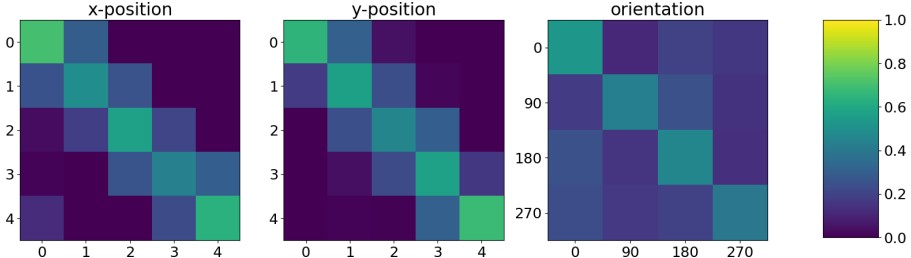

Figure 7: Confusion matrices between the values predicted by the perceptor (horizontally) and the true values (vertically) for each of the symbolic state components.

gradients algorithm, where we have replaced the program with a single linear layer with softmax output. Additionally, we rendered an arrow in the observed images to represent the chosen goal such that the policy network has access to this information as well. In only 2000 iterations, the perceptor obtains an average reward close to the optimal one of approximately 11.35, while it takes more than 30000 iterations for the policy gradients algorithm to approach an average reward of 10. Furthermore, given the highly structured nature of the environment and the task, the perceptor gradients agent eventually learns to solve the task with much greater reliability than the pure policy gradients one. See A.5 for experimental results on the Minecraft tasks with a feedforward perceptor instead of an autoencoding one.

### 5.2.2 PERCEPTOR LATENT SPACE

In this case the latent space directly maps to the symbolic state description of the environment and so we collected a dataset of 1000 episodes and compared the output of the perceptor to the ground truth values. The results, summarised in the confusion matrices in Figure 7, indicate that the perceptor has learnt to correctly ground the symbolic state description to the observed data. Despite the fact that there are some errors, especially for the orientation, the agent still learnt to perform the task reliably. This is the case because individual errors at a certain timestep have a limited impact on the eventual solution of the task due to the robust nature of the abstractions supported by the symbolic A* planner.

### 5.2.3 GENERATING OBSERVATIONS

Since we have trained an autoencoding perceptor we are also able to generate new observations, in this case images of the environment. Typically, manual inspection of the latent space is needed in order to assign each dimension to a meaningful quantity such as position or orientation. However, this is not the case for autoencoding perceptors, as the program attaches a semantic meaning to each of the symbols it consumes as an input. Therefore, we can generate observations according to a semantically meaningful symbolic specification without any inspection of the latent space. In Figure 8, we have chosen a sequence of symbolic state descriptions, forming a trajectory, that we have passed through the decoder in order to generate the corresponding images. Given the lack of noise in the rendering process we are able to reconstruct images from their symbolic descriptions almost perfectly. Nevertheless, some minor orientation related artefacts can be seen in the generated image corresponding to $t = 5$.

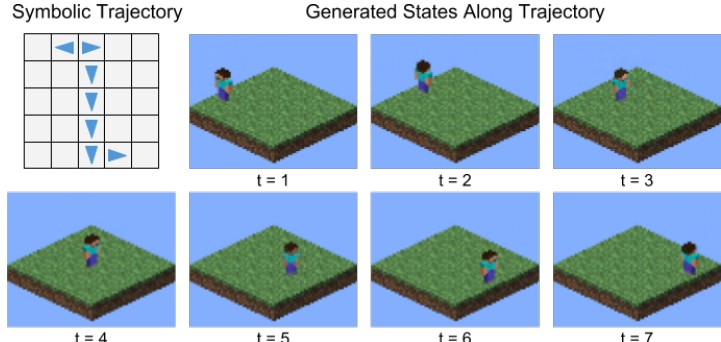

Figure 8: Sampled images of states following a symbolically defined trajectory.

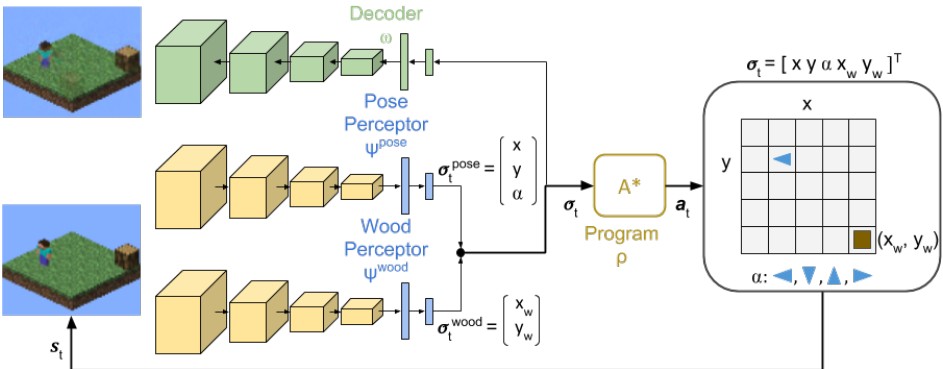

Figure 9: A diagram of the 'collect wood' experimental setup (baseline networks not illustrated).

## 5.3 MINECRAFT: COLLECT WOOD

**Task Description** The last task we consider is navigating to and picking up an item, in particular a block of wood, from the environment. In addition to the motion actions, the agent can now also pick up an item if it is directly in front of it. The pick action succeeds with 50% chance, thus, introducing stochasticity in the task. As discussed by Levesque (2005), this problem cannot be solved with a fixed length plan and requires a loop in order to guarantee successful completion of the task. The agent receives +5 reward whenever it picks up a block of wood. We expand the state to include the location of the wood block resulting in

$$\boldsymbol{\sigma} = \begin{bmatrix} x & y & \alpha & x_w & y_w \end{bmatrix}^T \tag{15}$$

where $x_w, y_w \in \{1, 2, 3, 4, 5\}$ are categorical variables representing the location of the block of wood and $x$, $y$, and $\alpha$ are defined as before. The symbolic state representation as well as the rendered observations are shown in Figure 9.

**Stacked Perceptors** In the cart-pole experiment the controller balances the system around a fixed state, whereas in the navigation experiment the A* planner takes the agent to a randomly chosen, but known, goal pose. Learning to recognise both the pose of the agent and the position of the item is an ill posed problem since there is not a fixed frame of reference – the same sequence of actions can be successfully applied to the same relative configuration of the initial state and the goal, which can appear anywhere in the environment. We have, however, already trained a perceptor to recognise the pose of the agent in the 'go to pose' task and so in this experiment we demonstrate how it can be transferred to a new task. We combine the pose perceptor from the previous experiment with another perceptor that is to learn to recognise the position of the wood block. Given the symbolic output of both perceptors, we can simply concatenate their outputs and feed them directly into the A* program, as shown in Figure 9. Even though the symbols of the pose perceptor are directly transferable to the current task, it needs to be adapted to the presence of the unseen before wooden block. We train both perceptors jointly, but keep the learning rate for the pre-trained pose perceptor

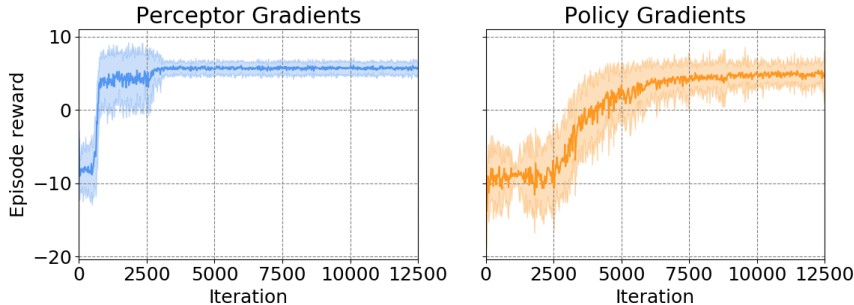

Figure 10: Learning performance at the 'collect wood' task of the perceptor gradients (left) compared to the policy gradients (right).

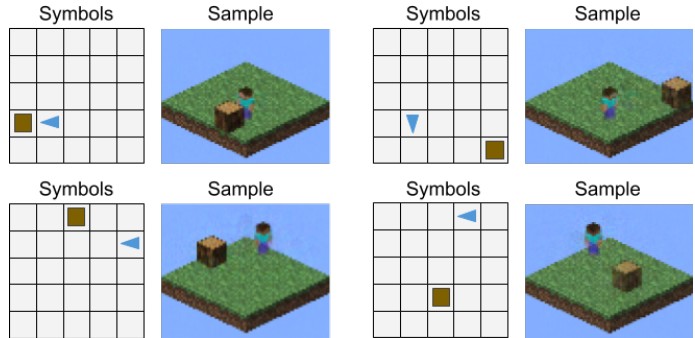

Figure 11: Sampled images from a symbolic specification over the joint latent space of the pose and wood perceptors.

considerably lower than the one for the wood perceptor. This ensures that no catastrophic forgetting occurs during the initial stages of training.

### 5.3.1 LEARNING PERFORMANCE

As shown in Figure 10 the agent is quickly able to solve the task with the stacked configuration of perceptors. It significantly outperforms the neural network policy that is trained on the entire problem and achieves optimal performance in less than 3000 iterations. These result clearly demonstrate that the symbolic representations learnt by a perceptor can be transferred to new tasks. Furthermore, perceptors not only can be reused, but also adapted to the new task in a lifelong learning fashion, which is reminiscent of the findings in (Gaunt et al., 2017) and their idea of neural libraries.

### 5.3.2 GENERATING OBSERVATIONS

Stacking perceptors preserves the symbolic nature of the latent space and so we are again able to generate observations from semantically meaningful specifications. In Figure 11 we have shown a set of generated state samples from a symbolic specification over the joint latent space. The samples not only look realistic, but also take occlusions into account correctly (e.g. see top left sample).

## 6 CONCLUSION

In this paper we introduced the perceptor gradients algorithm for learning programmatically structured representations. This is achieved by combining a perceptor neural network with a task encoding program. Our approach achieves faster learning rates compared to methods based solely on neural networks and yields transferable task related symbolic representations which also carry semantic content. Our results clearly demonstrate that programmatic regularisation is a general technique for structured representation learning.

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

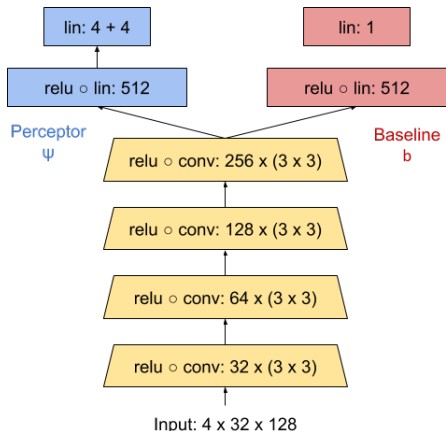

Figure 12: Architecture of the cart-pole feedforward perceptor $\psi_\theta$ and the baseline network $b_\phi$. Convolutions are represented as #filters $\times$ filter size and all of them have a stride of 2. Linear layers are denoted by the number of output units.

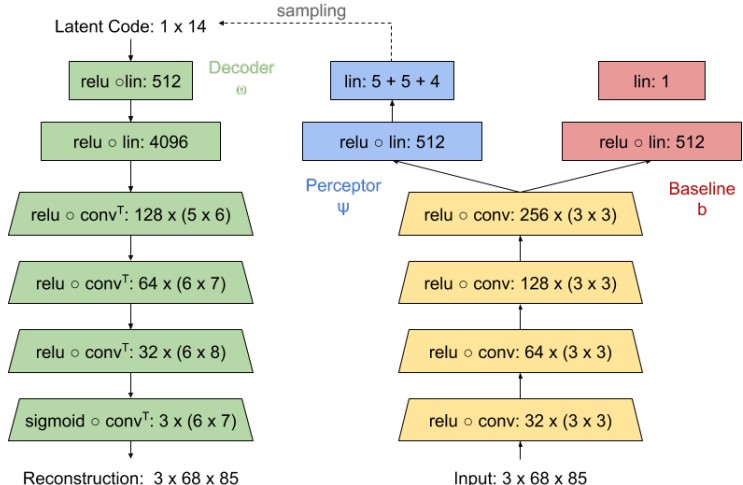

Figure 13: Architecture of the 'go-to pose' autoencoding perceptor $\psi_\theta$, the decoder $\omega_\upsilon$ and the baseline network $b_\phi$. Convolutions are represented as #filters $\times$ filter size and all of them have a stride of 2. Transposed convolutions ($conv^T$) are represented in the same way. Linear layers are denoted by the number of output units.

## A    SUPPLEMENTARY INFORMATION

### A.1    CART-POLE FEEDFORWARD PERCEPTOR

The input of the cart-pole feedforward perceptor is a stack of 4 consecutive grayscale $32 \times 128$ images that we render the cart-pole system onto as shown in Figure 2. This is a setup similar to the one proposed in (Mnih et al., 2015) which preserves temporary information in the input such that it can be processed by a convolutional neural network. The architecture of the perceptor $\psi_\theta$ is shown in Figure 12. Note that the perceptor shares its convolutional layers with the baseline network $b_\phi$. The outputs of the perceptor are the mean and the diagonal covariance matrix of a 4-dimensional normal distribution.

### A.2    MINECRAFT 'GO-TO POSE' AUTOENCODING PERCEPTOR

For this experiment we designed an autoencoding perceptor, the architecture of which is shown in Figure 13. The input is a single color image containing 2.5D rendering of the world as shown in

Figure 5. The encoder outputs the parameters of 3 categorical distributions corresponding to $x$, $y$ and $\alpha$ variables. These distributions are sampled to generate a latent code that is put through the decoder. We use the Gumbel-Softmax reparameterisation of the categorical distributions (Jang et al., 2016) such that gradients can flow from the decoder through the latent code to the encoder.

## A.3 Cart-Pole State Space Model

The state vector of the cart-pole system is

$$\boldsymbol{\sigma} = [x \quad \dot{x} \quad \alpha \quad \dot{\alpha}]^T \tag{16}$$

where $x \in \mathbb{R}$ is the linear position of the cart and $\alpha \in \mathbb{R}$ is the angle of the pendulum with respect to its vertical position as shown in Figure 2. By following the derivation in (Lam, 2004) of the linearised state space model of the system around the unstable equilibrium $[0\ 0\ 0\ 0]^T$ (we ignore the modelling of the gearbox and the motor) we set the system matrix $\boldsymbol{A}$ and input matrix $\boldsymbol{B}$ to

$$\boldsymbol{A} = \begin{bmatrix} 0 & 1 & 0 & 0 \\ 0 & 0 & -\frac{gml}{LM-ml} & 0 \\ 0 & 0 & 0 & 1 \\ 0 & 0 & \frac{g}{L-ml/M} & 0 \end{bmatrix} \quad \boldsymbol{B} = \begin{bmatrix} 0 \\ \frac{1}{M-ml/L} \\ 0 \\ -\frac{1}{ML-ml} \end{bmatrix} \tag{17}$$

where $m$ is the mass of the pole, $M$ is the mass of the pole and the cart, $l$ is half of the pendulum length, $g$ is the gravitational acceleration, $L$ is set to $\frac{I+ml^2}{ml}$ and $I = \frac{ml^2}{12}$ is the moment of inertia of the pendulum. We use the cart-pole system from OpenAI gym where all the parameters are pre-specified and set to $m = 0.1$, $M = 1.0$ and $l = 0.5$.

## A.4 LQR State Space Uniqueness

Given that $\boldsymbol{Q}$ is a matrix of the form $\lambda \boldsymbol{I}_4$ where $\lambda$ is a scalar, then any rotation matrix $\boldsymbol{M}$ applied on the latent vector $\boldsymbol{\sigma}$ will have no impact on the cost in (12) as

$$(\boldsymbol{M}\boldsymbol{\sigma})^T \boldsymbol{Q} \boldsymbol{M}\boldsymbol{\sigma} = \boldsymbol{\sigma}^T \boldsymbol{M}^T \lambda \boldsymbol{I}_4 \boldsymbol{M}\boldsymbol{\sigma} = \boldsymbol{\sigma}^T \lambda \boldsymbol{I}_4 \boldsymbol{M}^T \boldsymbol{M}\boldsymbol{\sigma} = \boldsymbol{\sigma}^T \lambda \boldsymbol{I}_4 \boldsymbol{\sigma} = \boldsymbol{\sigma}^T \boldsymbol{Q}\boldsymbol{\sigma} \tag{18}$$

since rotation matrices are orthogonal. Additionally, scaling $\boldsymbol{\sigma}$ only scales the cost function and so will not shift the locations of the optima.

## A.5 Minecraft Tasks with a Feedforward Perceptor

In order to study the contribution of the decoder to the performance of the agent in the Minecraft tasks we conducted a set of ablation experiments where we replaced the autoencoding perceptor with a feedforward one. Figure 14 and Figure 15 show the learning performance at the 'go-to pose' and 'collect-wood' tasks, respectively, with a feedforward perceptor. Overall, the results indicate that the main effect of the decoder is to decrease the variance of the obtained reward during training. The feedforward perceptor manages to ground the position of the agent slightly more accurately than the autoencoding perceptor, however the accuracy of the orientation has decreased. The reason for this is that orientation has little effect on the performance of the agent as it can move to any square around it, regardless of its heading. This is similar to the cart-pole task where the linear and angular velocities had little effect on the LQR performance.

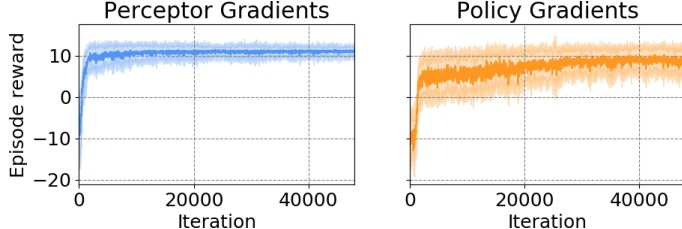

Figure 14: Learning performance with a feedforward perceptor at the 'go to pose' task of the perceptor gradients (left) compared to the policy gradients (right).

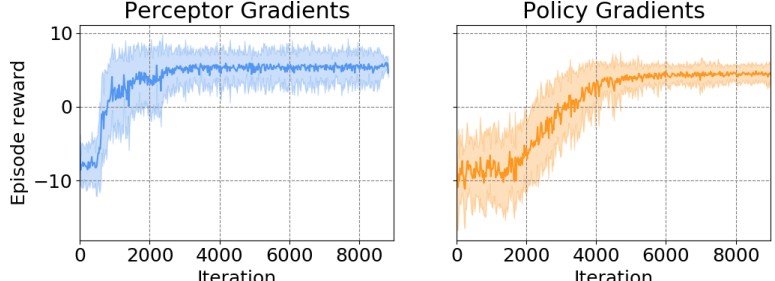

Figure 15: Learning performance with a feedforward perceptor at the 'collect wood' task of the perceptor gradients (left) compared to the policy gradients (right).

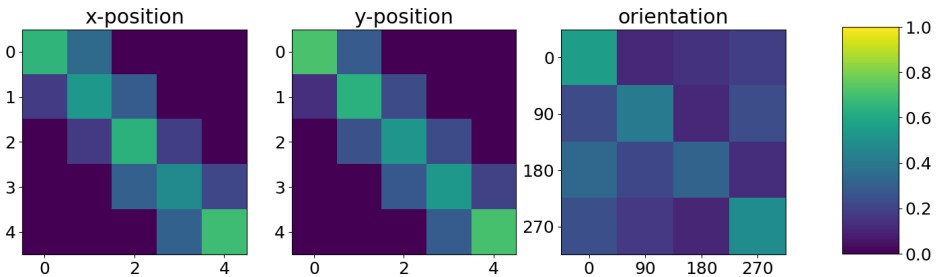

Figure 16: Confusion matrices between the values predicted by the feedforward perceptor (horizontally) and the true values (vertically) for each of the symbolic state components.

