# OpenReview forum: "Learning Programmatically Structured Representations with Perceptor Gradients"
_ICLR.cc/2019/Conference_

### Official Review · AnonReviewer3 · 2018-11-03
**I tried very hard but I think ultimately failed to understand this paper.**

**Rating:** 5
**Confidence:** 3

**Review:**

The fundamental idea proposed in this paper is a sensible one:  design the functional form of a policy so that there is an initial parameterized stage that operates on perceptual input and outputs some "symbolic" (I'd be happier if we could just call them "discrete") characterization of the input, and then an arbitrary program that operates on the symbolic output of the first stage.

My fundamental problem is with equation 3.  If you want to talk about the factoring of the probability distribution p(a | s) that's fine, but, to do it in fine detail, it should be:
P(a | s) = \sum_sigma P(a, sigma | s) = \sum_sigma P(a | sigma, s) * P(sigma | s)
And then by conditional independence of a from s given sigma
 = \sum_sigma P(a | sigma) * P(sigma | s)
But, critically, there needs to be a sum over sigma!  Now, it could be that I am misunderstanding your notation and you mean for p(a | sigma) to stand for a whole factor and for the operation in (3) to be factor multiplication, but I don't think that's what is going on.

Then, I think, you go on to assume, that p(a | sigma) is a delta distribution.  That's fine.

But then equation 5 in Theorem 1 again seems to mention delta without summing over it, which still seems incorrect to me.

And, ultimately, I think the theorem doesn't make sense because the transformation that the program performs on its input is not included in the gradient computation.  Consider the case where the program always outputs action 0 no matter what its symbolic input is.   Then the gradient of the log prob of a trajectory with respect to theta should be 0, but instead you end up with the gradient of the log prob of the symbol trajectory with respect to theta.

I got so hung up here that I didn't feel I could evaluate the rest of the paper.

One other point is that there is a lot of work that is closely related to this at the high level, including papers about Value Iteration Networks, QMDP Networks, Particle Filter Networks, etc.  They all combine a fixed program with a parametric part and differentiate the whole transformation to do gradient updates.  It would be important in any revision of this paper to connect with that literature.

---

> ### Comment · AnonReviewer4 · 2018-11-09
> **My impression of the purpose of Theorem 1**
>
> With respect to how the program's transformation is included in the gradient computation, my understanding from equation 8 is that the point of Theorem 1 is to show that, because the program is a non-differentiable piece, we can essentially push the agent/environment boundary further into the agent, such that the "actions" are the task related symbols, the "states" are the visual observations, and the "agent" is only the perceptor network. Then, from the perspective of the policy parameters, the program essentially becomes part of the environment. Therefore, we can apply REINFORCE to optimize the perceptor network as we would for any other policy.

---

> > ### Author Response · Authors · 2018-11-11
> > **Purpose of Theorem 1**
> >
> > This is indeed the purpose of Theorem 1 as we have also mentioned in our response to the reviewer.

---

> ### Author Response · Authors · 2018-11-11
> **Response to the review**
>
>
> Factorisation in Equation 3:
> ------------------------------------------
> Thank you for the close look at the mathematical details of the paper. Equation 3 is meant to represent full factor multiplication rather than marginalisation of \sigma_t. We will clarify this in the paper and hopefully avoid confusing other readers.
>
> Theorem 1:
> -----------------
> The purpose of Theorem 1 is to show that REINFORCE can be applied to train the perceptor network. The key intuition behind Theorem 1 is that the program can be thought of as being absorbed in the environment and the task of the agent is to feed it the right inputs. Therefore, considering a program that always outputs action 0, as suggested in the review, is essentially equivalent to an environment which does not take into account the actions of the agent at all. In this case, the gradient of the log probability of the trajectory with respect to the parameters of the policy, in a standard policy gradients setup, would also be non-zero. More importantly, while this scenario is an interesting theoretical edge case it has little, if any, practical implications.
>
> Additionally, we would like to note that Theorem 1 handles correctly the case when the perceptor is to always output the same symbols as the gradient of the log prob of the symbol trajectory with respect to theta will be 0 as expected.
>
> Related Work:
> ----------------------
> Works such as Value Iteration Networks, QMDP Networks and Particle Filter Networks are based on the idea of differentiable programs which can express only subset of the problems that a general program can express.
> One of the key contributions of the paper is that perceptor gradients can work with general programs as we have demonstrated by directly plugging in programs from standard Python packages. Nevertheless, that is a substantial body of literature that we will update the paper to connect with.
>
> Symbols vs. Discrete variables:
> -----------------------------------------------
> The perceptor can output both continuous (LQR experiment) and discrete variables (Minecraft experiments) that characterise the raw input data. We call the output of the perceptor symbolic as each output variable (regardless of its domain) has semantic content imposed by the program. Our experiments demonstrate that the perceptor does learn representations which follow the symbolic structure of the program.

---

> > ### Comment · AnonReviewer3 · 2018-11-11
> > **Now I see...**
> >
> > So, if equation (3) is a factor multiply, then we can write it out as:
> > \[\pi(a \mid s) =  \sum_\sigma P(a \mid \sigma) P(\sigma \mid s)\;\;,\]
> > which by your assumption of a deterministic program is
> > \begin{align*}
> > \pi(a \mid s) & =  \sum_\sigma I(\rho(\sigma) = a)  P(\sigma \mid s)\\
> > &  =  \sum_{\{\sigma \mid \rho(\sigma) = a\}}  \psi(\sigma \mid s)
> > \end{align*}
> > So
> > \begin{align}
> > \nabla_\theta \log \pi(a \mid s) &= \nabla_\theta \sum_{\{\sigma \mid \rho(\sigma) = a\}}  \psi(\sigma \mid s)\\
> > & = \sum_{\{\sigma \mid \rho(\sigma) = a\}}  \nabla_\theta \psi(\sigma \mid s)
> > \end{align}
> > Note that this quantity still depends on which symbols $\sigma$ will
> > cause your program $\rho$ to generate action $a$.
> >
> > Now, in equation (6), $\sigma_t^{(i)}$ a particular symbol, which is the one
> > that was {\em actually} generated by the perceptor at time $t$ on sequence
> > $i$, right?   Your equation (6) makes sense to  me if $\sigma$ is
> > actually part of $\tau$.  But it's not.
> >
> > Of course my example of a program that ignores its input and always
> > outputs 0 is not interesting practically.  But let's see what happens
> > here:
> > \begin{itemize}
> > \item The trace will only have $a = 0$.
> > \item The sequence of symbols $\sigma_t^{(i)}$ will be something
> >   interesting.
> > \item In your equation (6), the right-hand side would be independent
> >   of $\rho$ and generally be non-zero.
> > \item But, in fact, my equation (1) above would be 0 because $\phi$
> >   is a probability distribution, and so it should be 0.  And also
> >   because, intuitively, it should be 0.
> > \end{itemize}
> >
> > In your reply to me you said:  ``the key intuition behind Theorem 1 is
> > that the program can be thought of as being absorbed in the
> > environment.''  But, in that case, it really {\em is} true that the
> > $\sigma$ need to be observed, because they are the new ``actions.''
> >
> > Okay.  I think this is why we end up with different understandings of
> > what's going on.
> >
> > Let me proceed with the rest of the paper under that assumption (but
> > if that's in fact what's going on here, then you would need to amend
> > your description of $\tau$ to include $\sigma$.)
> >
> > Okay.  Sorry to have been dim.  It does all make sense now.
> >
> > This seems like a completely reasonable idea, though really not all
> > that surprising.  One might ask whether it would be a
> > good idea to do value-iteration networks this way, too:  that is,
> > think of the VIN as part of the environment and just train the models
> > using reinforce.  I guess not, quite, because the model for VIN is a
> > static object, not something that varies along the trajectory.
> >
> > Then we need to get into the question of whether reinforce is a
> > sensible algorithm or not, and under what circumstances.
> >
> > In any case, I will change my rating and hope the story of my
> > confusion above is useful to you.

---

### Official Review · AnonReviewer2 · 2018-11-07
**Learning Programmatically Structured Representations with Perceptor Gradients**

**Rating:** 6
**Confidence:** 1

**Review:**

This paper proposes the perceptor gradients algorithm to learn symbolic representations for devising autonomous agent policies to act.
The perceptor gradients algorithm decomposes a typical policy into a perceptor network that maps observations to symbolic representations and a user-provided task encoding program which is executed on the perceived symbols in order to generate an action. Experiments show the proposed approach achieves faster learning rates compared to methods based solely on neural networks and yields transferable task related symbolic representations. The results prove the programmatic regularisation is a general technique for structured representation learning. Although the reviewer is out of the area in this paper, this paper seems to propose a novel algorithm to learn the symbolic representations.

---

### Official Review · AnonReviewer4 · 2018-11-09
**Interesting perspective, but the paper could be stronger with experiments that reflect its original motivations**

**Rating:** 7
**Confidence:** 5

**Review:**

The high-level problem this paper tackles is that of learning symbolic representations from raw noisy data, based on the hypothesis that symbolic representations that are grounded in the semantic content of the environment are less susceptible to overfitting.

The authors propose the perceptor gradients algorithm, which decouples the policy into 1) a perceptor network that maps raw observations to domain-specific representations, which are inputs to 2) a pre-specified domain-specific control or planning program. The authors claim that such a decomposition is general enough to accommodate any task encoding program.

The proposed method is evaluated on three experiments: a simple control task (cartpole-balancing), a navigation task (minecraft: go to pose), and a stochastic single-object retrieval task (minecraft: collect wood). The authors show that the perceptor gradients algorithm learns much faster than vanilla policy gradient. They also show that the program provides an inductive bias that helps ground the representations to the true state of the agent by manually inspecting the representations and by reconstructing the representation into a semantically coherent scene.

This paper is clear and well-written and I enjoyed reading it. It proposes a nice perspective of leveraging programmatic domain knowledge and integrating such knowledge with a learned policy for planning and control. If the following concerns were addressed I would consider increasing my score.

1. To what extent do the experiments support the authors' claims: Although the existing experiments are very illustrative and clear, they did not seem to me to illustrate that the learned representations are transferable as the authors claimed in the introduction. This perhaps is due to the ambiguous definition of "transferable;" it would be helpful if the authors clarified what they mean by this. Nevertheless, as the paper suggests in the introduction that symbolic representations are less likely to overfit to the training distribution, I would be interested to see an experiment that illustrates the capability of the program-augmented policy to generalize to new tasks. For example, Ellis et al. [1] suggested that the programs can be leveraged to extrapolate to problems not previously seen in the input (e.g. by running the for loop for more iterations). To show the transferability of such symbolic representations, is it possible for the authors to include an experiment to show to what extent the perceptor gradients algorithm can generalize to new problems? For example, is it possible for the proposed approach to train on "Minecraft: Go to Pose" and generalize to a larger map? Or is it possible for the proposed approach to train on one wood block and generalize to more wood blocks?
2. Experiment request: The paper seems to suggest that the "Minecraft: Go to Pose" task and the "Minecraft: Collect Wood" task were trained with an autoencoding perceptor. To more completely assess to what extent the program is responsible for biasing the representations to be semantically grounded in environment, would the authors please provide ablation experiments (learning curves, and visualization of the representations) for these two tasks where only the encoder was used?
3. Question: I am a bit confused by the beta-VAE results in Figure 4. If the beta-VAE is trained to not only reconstruct its input as well as perform a "linear regression between the learnt latent space and the ground truth values" (page 6), then I would have expected that the latent space representations to match the ground truth values much more closely. Would the authors be able to elaborate more on the training details and objective function of the beta-VAE and provide an explanation for why the learned latent space deviates so far from the ground truth?
5. Related work:
    a) The paper briefly discusses representation learning in computer vision and physical dynamics modeling. However, in these same domains it lacks a discussion of approaches that do use programs to constrain learned representations, as in [1-3]. Without this discussion, my view is that the related work would be very incomplete because program-induced constraints are core to this paper. Can the authors please provide a more thorough and complete treatment of this area?
    b) The approaches that this paper discusses for representation learning have been around for quite a long time, but it seems rather a misrepresentation of the related work to have all but two citations in the Related Work section 2017 and after. For example, statistical constraints on the latent space have been explored in [4-5]. Can the authors please provide a more thorough and complete treatment of the related work?
6. Possible limitation: A potential limitation for decoupling the policy in this particular way is that if the perceptor network produced incorrect representations that are fed into the program, the program cannot compensate for these errors. It would be helpful for the authors to include a discussion about this in paper.
7. How does this scale? As stated in the intro, the motivation for this work is for enabling autonomous agents to learn from raw visual data. Though the experiments in this paper were illustrative of the approach, these experiments assumed that the agent had access to the true state variables of its environment (like position and velocity), and the perceptor network is just inferring the particular values of these variables for a particular problem instance. However, presumably the motivation for learning from raw visual data is that the agent does not have access to the simulator of the environment. How do the authors envision their proposed approach scaling to real world settings where the true state variables are unknown? There is currently not an experiment that shows a need for learning from raw visual data. This is a major concern, because if the only domains that the perceptor gradients algorithm can be applied are those where the agent already has access to the true state variables, then there may be no need to learn from pixels in the first place. This paper would be made significantly stronger with an experiment where 1) learning from raw visual data is necessary (for example, if it is the real world, or if the true state variables were unknown) and 2) where the inductive bias provided by the program helps significantly on that task in terms of learning and transfer. Such an experiment would decisively reflect the paper's claims.
8. Clarity: The paper mentions that it is possible to generate new observations from the latent space. The paper can be made stronger by a more motivated discussion of why generating new observations is desirable, beyond just as a visualization tool. For example, the authors may consider making a connection with the analysis-by-synthesis paradigm that characterizes the Helmholtz machine.

[1] Ellis et al. (https://arxiv.org/pdf/1707.09627.pdf)
[2] Wu et al. (http://papers.nips.cc/paper/6620-learning-to-see-physics-via-visual-de-animation.pdf)
[3] Kulkarni et al. (https://www.cv-foundation.org/openaccess/content_cvpr_2015/html/Kulkarni_Picture_A_Probabilistic_2015_CVPR_paper.html)
[4] Schmidhuber (ftp://ftp.idsia.ch/pub/juergen/factorial.pdf)
[5] Bengio et al. (https://arxiv.org/abs/1206.5538)

---

> ### Author Response · Authors · 2018-11-25
> **Response 1 / 2**
>
> Thank you very much for the detailed  review and the interesting points and suggestions made. We have split our response in two parts due to response length limits.
>
>   >> 1. To what extent do the experiments...
>
> There are two main aspects of transferability that we consider:
>
> 1) Transferability to new environments - can a perceptor network trained in one environment be transferred to another? Extrapolating with neural networks to new parts of the data domain that have not been considered during training is a hard challenge. We address this problem with the “Minecraft: Collect Wood” experiment where we used a pose perceptor trained on the “Minecraft: Go to Pose”. The pose perceptor network had never seen a wood block and so we allow for very slow adaptation of the pose perceptor during learning of the “Go to Pose” task. This demonstrates that perceptors can be transferred to new environments. Of course, methods such as data augmentation and domain randomisation can improve the transferability of perceptors, but ultimately they do inherit all the limitations of pure statistical learners.
>
> 2) Transferability to new tasks - can a perceptor trained on one task be used for another? The output of a perceptor, due to its symbolic nature, can be fed into a variety of programs solving different tasks in the same environment. For example, one can easily modify the LQR controller to stabilise the pendulum at any other linear position x different from 0. One can also modify the A* planner such that the agent avoids the wooden block, rather than collecting it. In contrast to the proposed perceptor-program decomposition, it is far from obvious how to alter an end-to-end policy in order to adhere to the specifications of a new task. The “Minecraft: Collect Wood” experiment addresses this idea to an extent by showing that a certain set of symbols, e.g. the agent pose, is required by many task encoding programs.
>
>   >> 2. Experiment request: …
>
> The program maps symbols to actions and so if it is removed then it needs to be replaced by neural network performing the same type of mapping. The main question then is what neural network architecture to use - if the architecture is too simple it will lead to poor performance and if it is too complex then it would result in learning poor symbolic representations.
> We have, however, revised the paper to include the results of the Minecraft experiments using a feedforward perceptor only (no decoder). The results demonstrate that the decoder has little effect on the learning performance and so the program introduces the main inductive bias. We would be happy to discuss other alternatives for such ablation experiments.
>
>   >> 3. Question: …
>
> Training of the beta-VAE is independent of the linear regression and these are performed sequentially. We use a vanilla beta-VAE as described in [1]. The resulting latent space is non-identifiable meaning that a certain factor of variation can be represented by any of the latent variables. In order to overcome this issue Higgins et al. [1] train a linear classifier on top of the learnt latent space in order to derive a disentanglement metric. We take a similar approach.  In order to inspect the latent space learnt by the beta-VAE we train a single layer linear regressor to predict the ground truth values from the latent code of the already trained beta-VAE.
>
> The key idea behind beta-VAE is that enforcing independence between the latent variables results in disentangled factors. This, however, is only the case when the ground truth factors of variation are indeed independent of each other. That assumption is obviously violated since the factors of variation are entangled through the physics model (in equations of motion written as an ODE) of the cart-pole system. Therefore, the beta-VAE does not manage to reconstruct the factors of variation as accurately as the perceptor gradients setup. Programmatic regularisation is a powerful technique precisely because it can be used to express and enforce arbitrary relationships between the latent factors of interest.
>
> [1] Higgins et al., beta-VAE: Learning Basic Visual Concepts with a Constrained Variational Framework, https://openreview.net/forum?id=Sy2fzU9gl

---

> > ### Author Response · Authors · 2018-11-25
> > **Response 2 / 2**
> >
> >
> > >> 5. Related work: …
> >
> > These are both fair points and we have taken them into account in the revised version of the paper.
> >
> >  >> 6. Possible limitation: …
> >
> > Certainly the perceptor outputs incorrect representations at the early stages of training, but the learning procedure manages to improve the perceptor until it outputs the correct representations. Space permitting, we will include a discussion on this potential limitation.
> >
> >   >> 7. How does this scale? …
> >
> > If the variables of interest are entirely unknown then it is not possible to provide a program which can map the output of the perceptor to an action. Writing any such a program specifies (as a working hypothesis, at least) the variables/symbols of interest which then the perceptor learns to infer. Importantly, expressing the program with symbols at the right abstraction level, balancing the capabilities of the perceptor and the semantics of the task, is crucial. For example, writing a controller program working with low-level pixel features is a daunting task, but writing a program using the position and velocity of the pendulum is a straightforward exercise for anyone familiar with control theory. Scaling up the method to a real-world task would mainly require improvements to the grounding capabilities of the perceptor. We do not report any experiments on physical robot setups, but we’d say that the difference between perceiving the position of a pendulum, for example, from a real physical system and the synthetic videos can be addressed by increasing the complexity of the perceptor network. In fact, our experiments demonstrate that programs provide strong inductive bias and speed up learning in a way which is which is crucial for RL based robot learning.
> >
> >   >> 8. Clarity
> >
> > We have expanded the background section to cover the ideas of inverse graphics and inverse physics, which are particular instantiations of the analysis by synthesis idea. In general, we find the idea of generating observations from symbolic representations interesting as it can be useful for performing some sort of enumerative testing of the trained RL agent, but this is work beyond the scope of this paper.

---

> > > ### Comment · AnonReviewer4 · 2018-11-27
> > > **Revision address some of my concerns**
> > >
> > > 1) With regards to transfer, it would be useful to see a comparison with a purely neural network baseline: one can imagine pre-training the neural network baseline on the “go-to-pose” task, and then fine-tune the network in the “collect-wood” task. This may be a fairer comparison.
> > > 2) The authors’ reasoning makes sense, and I believe that this is a strength of the preceptor gradients framework.
> > > 3) I see. Thank you for clarifying.
> > > 7) Agreed, expressing the program at the right abstraction level would be crucial. Although writing a controller program whose inputs live in a relatively simpler abstraction level is easier, a major challenge I potentially see with how the perceptor gradients would scale. For example, in the Minecraft tasks, the outputs of the preceptor network are categorical variables over the agent’s x and y position. However, presumably for a task with this simplicity it may be possible to use conventional non-learning computer vision methods to obtain the x and y position. Beyond merely increasing the complexity of the preceptor network, there seems to be a conceptual issue: how would the perceptor gradients approach scale to scenarios where the state is not that easy to symbolically specify? This is presumably the motivation for learning from pixels in the first place. For example, one can imagine manipulating a non-convex object like a cup, or a soft object like a stuffed animal. Would the output of the perceptor network be in these cases, and would the output space of the perceptor network need to be custom-designed for each differing object geometry? I think the perceptor gradients approach is a good step towards learning systems that generalize better, but it seems that future work would need to address the challenge of scaling the approach to domains where the output space of the perceptor network is more complex than categorical positions.
> > >
> > > Overall, the revised version of the paper has addressed some of my concerns, although it still seems to me that more future work would need to be done with respect to point (7) for the perceptor gradients approach to have more impact. Despite these concerns, I believe the paper is a good first step towards tackling an ambitious goal, so I would recommend acceptance.

---

### Author Response · Authors · 2018-11-25
**Revised Paper**


We would like to thank the reviewers for all the feedback you have provided us as we feel that it has significantly improved our paper. We have uploaded a revised version of the paper taking into account all comments and suggestions. Key changes which we have made include:

- Expanded background section
- Improved the mathematical description to avoid confusion
- Included results from ablation experiments on the Minecraft tasks

---

### Meta-Review · Area_Chair1 · 2018-12-14

**Confidence:** 3
**Recommendation:** Accept (Poster)

**Metareview:**

This paper considers the problem of learning symbolic representations from raw data. The reviewers are split on the importance of the paper. The main argument in favor of acceptance is that bridges neural and symbolic approaches in the reinforcement learning problem domain, whereas most previous work that have attempted to bridge this gap have been in inverse graphics or physical dynamics settings. Hence, it makes for a contribution that is relevant to the ICLR community. The main downside is that the paper does not provide particularly surprising insights, and could become much stronger with more complex experimental domains.
It seems like the benefits slightly outweigh the weaknesses. Hence, I recommend accept.